# Functionalized Biscuits with Bioactive Ingredients Obtained by Citrus Lemon Pomace

**DOI:** 10.3390/foods10102460

**Published:** 2021-10-15

**Authors:** Valeria Imeneo, Rosa Romeo, Antonio Gattuso, Alessandra De Bruno, Amalia Piscopo

**Affiliations:** Department of AGRARIA, University Mediterranea of Reggio Calabria, 89124 Reggio Calabria, Italy; valeria.imeneo@unirc.it (V.I.); rosa.romeo@unirc.it (R.R.); antonio.gattuso@unirc.it (A.G.); amalia.piscopo@unirc.it (A.P.)

**Keywords:** antioxidant, biscuits, lemon pomace, phenolic compounds, oxidative stability, Ultra-high-pressure liquid chromatography (UHPLC)

## Abstract

In this study, functionalized biscuits were prepared through the enrichment of dough with lemon peel and natural antioxidants extracted from lemon pomace. Lemon pomace extract (LP_E_) was analyzed for total phenolic content before addition, and then a known concentration of 50 mg kg^−1^ was used for the formulation of enriched biscuits. Three different biscuit samples were compared to a control biscuit, without the addition of functional ingredients. The main physicochemical, microbiological, and sensory aspects of doughs and biscuits enriched with LP_E_ were investigated. The enriched biscuits showed higher phenolic content and antioxidant activity than the control one and a longer induction period (IP), which means that the enriched products had a higher intrinsic resistance to lipid oxidation, thanks to the antioxidant effect exerted by the added fresh lemon peel and the LP_E_. Furthermore, from a sensorial point of view, they showed suitable acceptability, in terms of appearance, flavor, and aromatic attributes. Thus, results indicated that the incorporation of lemon processing by-products allowed the production of functional enriched biscuits with improved antioxidant properties.

## 1. Introduction

Wastes are widely generated by food production, of whom fruits and vegetables have the highest waste rates of any food. Although these by-products still contain nutrients and bioactive compounds, they are considered a problem for their environmental footprint [1]. For this reason, the demand for more sustainable practices toward the application of a circular economy in the food system represents a key strategy for the future. Consequently, most studies are focused on improving the possibility to extract nutrients from by-products and using parts of them in the production of new functional foods [2,3,4,5]. In this context, citrus is one of the most important fruit crops in the world with an annual production exceeding 122.5 million tons, and one-third of the crop is processed, e.g., to produce juices, jams, or for the extraction of essential oils. Worldwide industrial citrus residues represent about 50% of the whole fruit mass: during citrus processing, peel residues are the primary waste fraction, amounting to almost half of the total weight of the fruit [6].

Particularly, lemon (*Citrus limon* (L.) Osbeck) pomace, a by-product of juice and essential oil processing industries, is a suitable source of sugars, minerals, organic acids, dietary fibers, and phenolic compounds, such as phenolic acids (ferulic, *p*-coumaric and sinapic acids) and flavonoids (flavanones, flavonols, flavones) with salubrious characteristics, such as antioxidants, anti-inflammatory and antimicrobial properties [7,8,9].

Particularly, lemon (*Citrus limon* (L.) Osbeck) pomace, a by-product of juice and essential oil processing industries, was reported to contain a series of bioactive components with salubrious characteristics, such as antioxidants, anti-inflammatory, and antimicrobial properties In response to consumers’ requests for healthier foods, the use of new functional ingredients recovered from food by-products could be a valid alternative for the formulation of high added value products [10]. Among bakery goods, biscuits could represent a viable and acceptable candidate for the addition of functional ingredients [11], even if their consumption is usually associated with an unhealthy lifestyle because of high levels of fat and sugar. They are the most consumed bakery products in many parts of the world thanks to their ready-to-eat nature, economic cost, nice interesting nutritional qualities, differentiation of tastes, and longer shelf life [12]. Much research has focused on the production of biscuits with functional ingredients from by-products [11,13,14,15]. At the same time, sensory quality is a fundamental aspect of most food products since it relates to their different taste perceptions, stability, and nutritional properties. Sensory evaluation is a scientific topic used to evoke, measure, analyze and interpret reactions to those characteristics of foods and materials as they are perceived by the senses of sight, smell, taste, touch, and hearing [16].

The aim of this research consists of quality evaluating of biscuits enriched with functional ingredients obtained from lemon by-products. Their physicochemical parameters, antioxidant profile, and sensory characters were discussed in comparison with the control ones to attest a higher quality from the healthy point of view.

## 2. Materials and Methods

### 2.1. Raw Material

Citrus lemon pomace (*Citrus limon* (L.) Osbeck) was supplied by Agrumaria Reggina company located in Gallico (Reggio Calabria, Italy) that produces juice and essential oils, and transported to the Food Technology laboratory of the Mediterranea University of Reggio Calabria where it was immediately freeze-dried (−70 °C) in a VirTis lyophilizer (SPscientific, Gardiner, NY, USA) and then vacuum-stored in polyethylene bags until subsequent. Wheat flour, corn seed oil, white sugar, cow skimmed milk, and chemical baking powder (disodium diphosphate, sodium hydrogen carbonate, cornstarch) were purchased into a supermarket to be used as ingredients for biscuit’s preparation.

### 2.2. Preparation of LP_E_

The extraction of antioxidant compounds from freeze-dried lemon pomace (LP) has been carried out according to Papoutsis et al. [17]. A total of 200 mL of ethanol:water (1:1, v:v) solution were mixed to 20 g ground LP in a Sonoplus Ultrasonic homogenizer, Series 2000.2, HD 2200.2 (BANDELIN, Ultraschall seit 1955), for 40 min at 40 °C. Subsequently, the sample was centrifuged (5000 rpm, 5 min, 4 °C, in a refrigerated centrifuge, (NF 1200R, Nüve, Ankara, Turkey), filtered (0.45 µm filter paper), and the resulting extract was diluted to 200 mL with the extraction solvent (food-grade ethanol:water, 1:1, v:v), which is reported to be suitable for the production of extracts to apply in the food system [18,19,20,21]. Before being used for the enrichment of biscuits, the extract thus obtained LP_E_ was subjected to qualitative and quantitative analysis.

### 2.3. Formulation of Enriched Biscuits with Bioactive Compounds

All the biscuit dough formulations and denominations are reported in Table 1. Samples were prepared to offer three variations compared to a control sample (A). The modifications to this formulation (A) were made to produce biscuits with the addition of: fresh lemon peel (B); fresh lemon peel and LP_E_ (C), and with only LP_E_ (D), by skimmed milk substitution. Concerning the preparation, corn seed oil and white sugar were firstly mixed (3200 g. min^−1^, 1 min) in an electric lab-scale mixer (Bimby TM31, Vorwerk, Wuppertal, Germany); then, the skimmed milk was added and mixed for two minutes (1800 g. min^−1^). Finally, the wheat flour and chemical baking powder were added and mixed (1100 g. min^−1^) for two more minutes. The obtained dough was rolled out with a calibrated rolling pin and shaped using a cutter for biscuits to assure the same thickness of approximately 3.5 mm for all the samples. The dough shapes were placed on a perforated rectangular tray and baked in a preheated industrial oven (Angelo Po Combistar FX, Carpi, Modena, Italy) for 7 min at 180 °C (Figure 1, example of final products). Subsequent product characterization analyses were carried out on both doughs and final products.

### 2.4. Characterization of Physicochemical Properties of LP_E_, Doughs, and Biscuits

#### 2.4.1. Physicochemical Evaluation

The moisture content in both doughs and baked biscuits was determined using a Sartorius Moisture Analyzer MA37 thermal balance by the gravimetric method at 105 °C until constant weight. The results were expressed as a percentage of moisture (%).

Water activity (a_w_) of doughs and biscuits was conducted using an hygrometer (Aqualab LITE, Decagon, Nelson Court, Pullman, Washington) and the pH of LP_E_ doughs and biscuits in a Crison pH-meter, basic model 20, according to the AOAC International Method [22].

The color analysis was performed using an automatic Minolta CR 300 tristimulus colorimeter. The CIE L*a*b* system was used as reference: the color parameters were L*, brightness (from 0 black to 100 white), a* (from –50 green to 50 red), b* (from −50 blue to 50 yellow), Chroma (C*), which represents the degree of saturation/fullness of color and it is calculated as (a^2^ + b^2^)^1/2^ and hue angle (H), which describes the amounts of redness and yellowness (from 90 yellow to 180 green), calculated as arctan (b*/a*). The measurement was carried out on 15 mL of LP_E_ in ten different points, placing the sample in an optical glass and directly on doughs and biscuits, performing the readings at different points.

#### 2.4.2. Microbiological Analysis

10 g of each dough and biscuit was homogenized for 3 min with 10 mL of Ringer solution using a Stomacher BagMixer 400 (Interscience, 30 Ch.Bois Arpents F.78860 St. Nom, France). Series dilutions of the homogenates were poured into the Petri plates in specific agar for total bacterial count (Plant Count Agar, Oxoid, at 25 ± 2 °C for 48 h), and for yeasts and molds count into the DRBC (Dichloran Rose Bengal Chloramphenicol) agar base plates (at 28 ± 2 °C for 48 h). At the end of the incubation period, the microbial colonies obtained were counted and expressed in Log10 colony-forming unit (CFU) g^−1^ of biscuit.

#### 2.4.3. Evaluation of Total Phenol Content and Antioxidant Activity

The phenolic extraction method for dough and biscuit samples was carried out following Miskiewicz et al. [23], with appropriate modifications. A total of 5 g of sample were ground in an electric grinder and mixed with 20 mL of methanol, 2.5 mL of distilled water, and 0.250 mL of concentrated hydrochloric acid (HCl). The mixtures were placed in a Sonoplus ultrasonic bath (Series 2000.2, HD 2200.2 (BANDELIN, Berlin, Germania) for 60 min at 30 °C and with a frequency of 20 kHz ± 500 Hz. After the extract was centrifuged for 10 min at 6000 rpm at 4 °C. The supernatant of the respective formulations was recovered, filtered (through a Whatman n. 4 filter), and made up to volume in a 25 mL flask with a 1:10 methanol: water mixture.

##### Total Phenolic Compounds (TPC)

TPC was determined according to the method reported by González–Molina et al. [24], with appropriate modifications. For the reaction, different concentrations of the extract were used, or rather: 0.2 mL of LP_E_ and 1 mL of doughs and biscuits extracts, that were placed inside a 25 mL flask and mixed with 5 mL of deionized water and 1 mL of Folin–Ciocalteu reagent. After 8 min, 10 mL of saturated sodium carbonate solution (Na_2_CO_3_, 20%) were added and diluted to volume with deionized water. The mixtures were incubated for two hours at room temperature in the dark. The absorbance was measured at 765 nm against a blank using a double-beam ultraviolet-visible spectrophotometer (Perkin-Elmer UV-Vis λ2, Waltham, MA, USA) and compared with a gallic acid calibration curve (concentration between 1 and 10 mg kg^−1^). The results were expressed as mg of gallic acid equivalents g^−1^ dry weight of lemon pomace (mg of GAE g^−1^ d.w.) and as mg of gallic acid equivalents 100 g^−1^ dry weight of dough and biscuit (mg GA 100 g^−1^ d.w.).

##### Total Flavonoid Content (TF)

The total flavonoid content (TF) was quantified following the method described by Papoutsis et al. [17], with some modifications.

In brief, 0.5 mL of LP_E_, 1 mL of deionized water, and 0.15 mL of NaNO_2_ (5%, *w*/*v*) were placed in a 5 mL volumetric flask and incubated at room temperature for 6 min. Subsequently, 0.15 mL of AlCl_3_ (10%, *w*/*v*) was added and incubated at room temperature for 6 min. Then, 2 mL of NaOH (4%, *w*/*v*) was added, and lastly, 1.2 mL of deionized water was used to make up to volume. At the same time, a solution used as a blank was prepared with the same amounts of reagents but without the addition of the sample. The mixture was incubated in the dark for 15 min. The absorbance was measured at 510 nm against a blank using a double-beam ultraviolet-visible spectrophotometer (Perkin-Elmer UV-Vis λ2, Waltham, Massachusetts, USA) and compared to a catechin calibration curve (concentration between 1 and 50 mg kg^−1^). The results were expressed as mg of catechin equivalents g^−1^ dry weight of lemon pomace (mg CE g^−1^ d.w.).

##### DPPH Assay

The DPPH assay was performed as reported by Brand-Williams et al. [25], based on the reaction between the DPPH (2,2-diphenyl-1-picrylhydrazyl) radical and antioxidant compounds of the samples, resulting in discoloration of the reaction solution due to the extinction of the radical. In a cuvette, after appropriate dilution, 15 µL of sample (LP_E_, dough, and biscuit extracts) and 2985 µL of a 6 × 10^−5^ M of methanol solution of DPPH, were allowed to react under darkness for 30 min at room temperature. The absorbance was measured at 515 nm, against methanol as blank, using a double-beam ultraviolet-visible spectrophotometer (Perkin-Elmer UV-Vis λ2, Waltham, Massachusetts, USA) and compared with a Trolox calibration curve (from 3 to 18 μM). The results were expressed as µM Trolox g^−1^ dry weight of lemon pomace for LP_E_ (µM TE g^−1^ d.w.) and as µmol Trolox 100 g^−1^ dry weight of dough and biscuit (µmol TE 100 g^−1^ d.w.).

##### ABTS Assay

The antioxidant activity of LP_E_ was determined by ABTS (2,2′-azino-bis acid (3-ethylbenzothiazolin-6-sulfonic acid) assay, a spectrophotometric discoloration method [26]. The working solution was prepared by mixing two stock solutions of 7 mM ABTS solution and 2.4 mM potassium persulphate (K_2_S_2_O_8_) solution and was incubated at room temperature for 12 h in the dark to achieve a stable value of absorbance: the reaction between ABTS^+^ and potassium persulphate determines the direct production of a blue/green chromogen. The resulting ABTS^+^ solution was diluted with ethanol, showing an absorbance of 0.70 (±0.02) at 734 nm.

The reaction mixture was prepared by mixing 15 µL of LP_E_ and 2985 µL of the ethanol solution of ABTS^+^. The absorbance was measured after 6 min in the dark using a double-beam ultraviolet-visible spectrophotometer (Perkin-Elmer UV-Vis λ2, Waltham, Massachusetts, USA). The quenching of initial absorbance was plotted against the Trolox concentration (from 3 to 18 μM), and the results were expressed as µmol Trolox g^−1^ dry weight of lemon pomace (µmol TE g^−1^ d.w.).

#### 2.4.4. Identification and Quantification of Antioxidant Compounds

The individual antioxidant compounds were identified and quantified in LP_E_, dough, and biscuit following the methods reported by Romeo et al. [27], making the appropriate modifications. The chromatographic system comprised of a UHPLC PLATINblue (Knauer, Berlin, Germany) provided with a binary pump system, which uses a Knauer blue orchid column C18 (1.8 µm, 100 × 2 mm) coupled with a PDA–1 (Photo Diode Array Detector) PLATINblue (Knauer, Berlin, Germany). The Clarity 6.2 software was employed.

Extracts were filtered through a 0.22 μm nylon syringe filter (diameter 13 mm), and then 5 μL was injected into the system. The mobile phases were water acidified with acetic acid (pH 3.10) (A) and acetonitrile (B); the gradient elution program consisted of 0–3 min, 95% A and 5% B; 3–15 min, 95–60% A and 5–40% B; 15–15.5 min, 60–0% A and 40–100% B. Ultimately, restoration of the initial conditions was reached during analysis maintaining the column at 30 °C and the injection volume at 5 μL. Peaks were revealed at 280 nm. For the identification and quantification of each compound, external standards (concentration between 1 and 100 mg kg^−1^) were used, and results were expressed as mg 100 g^−1^ dry weight of lemon pomace and mg 100 g^−1^ dry weight of dough or biscuit.

#### 2.4.5. Oxidative Stability Study

The Oxitest (VELP Scientifica, Milano, Italy) method consists of monitoring the oxygen uptake by reactive components in the tested food samples, allowing evaluation of oxidative stability under accelerated oxidation conditions [5].

In this study, the analysis was performed on 30 g of minced biscuits distributed homogenously in a hermetically sealed titanium chamber and pressurized until 6 bars at 90 °C. Each accelerated oxidation test was repeated using a single reactor for a total of two analytical replies. The OXITEST (Oxidation Test Reactor) result is the induction period (IP), expressed as “stability time” before fat oxidation and corresponding to a decrease in O_2_ pressure due to the consumption of oxygen by the sample [28]. OXISoft™ Software (Version 10002948 Usmate Velate, MB, Italy) automatically calculates the IP (h).

#### 2.4.6. Sensory Analysis

The evaluation of sensory analysis was conducted to assess differences among the different samples through a preference test useful to allow comparisons between the control and the functionalized samples. The biscuits were scored compared to several qualitative factors, such as: appearance attributes (superficial brown color, firmness, crunchiness, friability, and fragrance), aromatic aspects (lemon, bergamot, caramel, and ethanol), and flavor (bitter and sweet). The test was performed by a panel of 10 judges (males and females) from 25 to 60 years old, recruited among researchers and technicians of the Food Science and Technology Unit of University Mediterranea of Reggio Calabria with previous experience in sensory analysis. The judges were trained before the sessions to identify the gustatory attributes to be evaluated. Sensory data were elaborated by calculating the median of results.

### 2.5. Statistical Analysis

In this study, each analysis was conducted in triplicate (n = 3), and all the experimental results were expressed as mean value ± standard deviation. The significant differences (*p* < 0.05) among mean values were determined by one-way analysis (ANOVA, Analysis of Variance), applying SPSS Software (Version 15.0, SPSS Inc., Chicago, IL, USA). A series of multiple comparisons, with Tukey’s post-hoc test, was performed to determine individual significant differences (*p* < 0.05). The Pearson’s correlation test was employed for the determination of correlation coefficients (r) among the extracted polyphenolic compounds and antioxidant assays.

## 3. Results and Discussion

### 3.1. Characterization of Lemon Pomace Extract (LP_E_)

Results of LP_E_ characterization are reported in Table 2. TPC and TF determinations denoted mean values of 0.59 ± 0.01 mg GA g^−1^ d.w. and 0.16 ± 0.00 mg CE g^−1^ d.w. respectively, with the predominant compounds represented by eriocitrin and hesperidin, as confirmed by literature [29]. The showed antioxidant activity was 36.29 ± 4.62 µmol TE g^−1^ d.w. The obtained results agreed with the ranges reported in the literature [30,31].

### 3.2. Physicochemical Analysis of Doughs and Biscuits

Although the water activity (a_w_) in the doughs was quite similar in all samples without any significant difference (Table 3), it was found that the a_w_ detected in shortcrust biscuit samples decreased significantly (*p* < 0.01). Shortcrust biscuits including only LP_E_ (sample D) had the lowest water activity value (a_w_ = 0.23 ± 0.01) of the other samples. This agrees with Miskiewicz et al. [23], who state that the water-binding capacity of the dough ingredients improved in the presence of the extract, reducing water evaporation in the process of baking.

The moisture percentages quantified in dough samples were similar among samples with the only exception of dough D, which showed the lowest value (*p* < 0.05) of moisture (14%) (Table 3). This result might be due to the different formulation that involved the only use of LP_E_ diluted in hydroalcoholic solution for the enrichment where ethanol possesses higher volatility than water. After the cooking, no significant differences in moisture were observed among the samples.

In both dough and shortcrust biscuit samples, the pH value was significantly lower (*p* < 0.01) in the products C and D, at 6.91 ± 0.11 and 6.87 ± 0.02 for enriched dough and 6.79 ± 0.02 and 6.72 ± 0.03 in biscuits, respectively (Table 3). This also suggests the possible influence of the LP_E_ on the presence of acidic components.

Concerning the color evaluation of dough samples (Table 4), color brightness (L*) did not vary significantly, whereas bigger variations (*p* < 0.01) were observed for the other chromatic components, especially for b* that increased in B and C, linked to the presence of fresh lemon peel in the formulation.

In bakery products, the final color is a result of chemical reactions, such as the Maillard reaction, which consists of non-enzymatic browning reactions affected by the quantity of reducing sugars in the foodstuff and by baking temperature [32,33] and caramelization [34]. As expected, comparing the colorimetric parameters measured for the dough and biscuit samples, a reduction in the brightness (L*) and a parallel increase in the values of components a* and b* were detected, being influenced by Maillard and caramelization reactions that occur during the baking process [10].

As found by Borrelli et al. [35], the reaction occurring between proteins and carbohydrates may be the cause of the final brown color: in this context, the superficial color is an important quality element, influencing the acceptability of baked goods formulated using wheat [15]. Even if baking parameters might affect baked food’s color because of also added ingredients [36], in this experimentation, no significant color difference was found among biscuit samples, denoting no visual influence by the LP_E_.

### 3.3. Microbiological Analysis

Results of microbiological analysis denoted a microbial growth only in dough samples, as showed in Table 5. The total bacterial count was similar among samples, with 2.3 Log10 UFC g^−1^, except for D, in which a lower count was enumerated (1.95 Log10 UFC g^−1^). This result might be due to the possible influence of the LP_E_ to reduce the bacterial growth, as confirmed by Ahmed and Noor [37] for citrus lemon extract and by Schieber et al. [38] for the antimicrobial and antioxidant activity of eriocitrin and hesperidin present in LP_E_.

The determination of mold and yeast colonies denoted quite similar counts among dough samples, around 1.5 Log10 UFC g^−1^. After the baking process, no total microbial, mold, and yeast counts were observed in the biscuits.

### 3.4. Total Polyphenol Content and Antioxidant Activity

Total polyphenol content (TPC) and antioxidant activity detected in dough and biscuit samples are reported in Figure 2.

The TPC values tended to double after the enrichment both in dough and biscuit samples (about 28 and 30 mg GA g^−1^ d.w. of dough and biscuit, respectively) compared to the control ones (about 12 and 15 mg GAE(Gallic acid equivalent) 100 g^−1^ d.w. of dough and biscuit, respectively). The addition of functional compounds to biscuit samples (B, C, and D), besides increasing the antioxidant properties in doughs, allows protecting the samples during the thermal treatment of cooking. Indeed, some of the valuable compounds initially added to the doughs were well preserved until the end of the baking process (*p* < 0.05). No significant differences were observed among B, C, and D samples (*p* > 0.05).

In this regard, as reported by Pasqualone et al. [10], the literature is not univocal about the effects of baking on phenolics that seem to depend on several factors, such as the type and structure of food ingredients and food matrix, source, and nature of bioactive compounds, recipe, and thermal processing [39,40,41,42]. Moreover, the increase in TPC of the biscuits may be due to the formation of neo-antioxidants compounds during the baking process because of the Maillard reaction [43]. As discussed for TPC, the enriched samples also showed significantly higher values of antioxidant activity by DPPH assay compared to the control ones. A positive correlation was observed between antioxidant activity and TPC both for doughs and biscuits, with Pearson’s correlation coefficients of 0.98 and 0.88, respectively. The antioxidant activity observed in control biscuits might be due to the presence of phenolic acids, which are the main antioxidants in wheat, and to their ability to scavenge free radicals [44].

### 3.5. Oxidative Stability Results

The oxidative reaction is particularly important in foods such as biscuits for the lipid content in their formulation [21]. The presence of antioxidant compounds in formulation preserves bakery goods for a longer time, resulting in an exceptional antioxidant effect [45]. Specifically, literature reports the lemon antioxidant activity, protecting the fatty component of the matrix during reactions of oxidation and influencing the formation of primary compounds, such as peroxides [28].

The indication of biscuits oxidation degree, represented by the induction period (IP), was obtained by the point of intersection between the line passing through the inflection of the curve and the inflection itself (Figure 3). This point corresponded to the end of the product’s intrinsic resistance to lipid oxidation and the beginning of the accelerated absorption of oxygen by the food (appearance of olfactory rancidity).

The results suggested that the enriched biscuits showed higher resistance to lipid oxidation compared to the control biscuit A (IP of 47 h). In particular, the highest IP (57 h) was recorded for sample C: it was reported that the synergistic antioxidant effect exerted by the added fresh lemon peel and the hydroalcoholic extract (LP_E_) contributed to preserving the biscuit from oxidation.

Indeed, as reported by Palombini et al. [46], the correlation between the presence of antioxidant compounds and the result of the accelerated oxidation test observed in enriched cookies could justify the connection between the detected antioxidant compounds in fresh lemon peel and extract and their possible action in protecting lipids against the oxidation process.

The same ingredients were also singularly active in B and D samples, with a slightly lower and similar IP (51 h). It was interesting to note that significantly higher values of TPC and antioxidant activity of C biscuit (Figure 2) were matched by the longest induction period (IP) observed in Figure 3.

### 3.6. UHPLC Phenolic Profile

The principal identified and quantified phenolic compounds in the enriched samples are reported in Table 6. C and B doughs possessed the highest amount of eriocitrin among the enriched samples, but only C dough showed the highest hesperidin value, as well as for both identified flavonoids in the enriched biscuits. From dough to biscuit, the specific amount tended to decrease significantly for eriocitrin in sample C and for hesperidin in sample D, probably due to the applied thermal process, and this effect was particularly evident in B biscuit for the absence of eriocitrin and hesperidin.

Comparing the amount of eriocitrin and hesperidin in baked biscuits, also in this case, it is evident that the richest is the one with fresh lemon peel and extract inside (C), followed by biscuit enriched only with the extract (D). At the same time, the biscuit formulated only with fresh lemon peel (B) did not show a determinable quantity of the two flavonoids identified in the other samples. This confirms the potential of enriched products to boost the day-to-day consumption of valuable compounds since biscuits are frequently consumed foods. The samples after the thermal treatment exhibited a significantly lower antioxidant activity (*p* < 0.05) than what was found in corresponding dough samples. 

### 3.7. Sensory Studies

Biscuits were also evaluated in terms of their sensory attributes, which are presented in Figure 4.

In all examined samples, the crust color was from pale to brownish yellow. The superficial brown color of the biscuits had lower scores with the addition of fresh lemon peel and LP_E_ (sample C), probably due to the lower amount of skimmed milk, thus less proteins and lactose, thus less Maillard reaction and caramelization. Indeed, the color and external aspect of biscuits are affected by reducing sugars, which caramelize during the baking process producing brown color [47]. The fragrance of the products was quite similar among biscuits, except for sample D. Moreover, D biscuits prepared by adding the LP_E_ and with the lowest a_w_ value (Table 4) was the crunchiest, confirming what discussed by Arimi et al. [48] about the water activity value, whose increase is associated with the plasticization of the matrix due to the higher amount of water, causing the product to lose its crispness, leading to softening of the matrix. The other appearance attributes are not significantly different between the control cookie and the enriched ones. The biscuit texture was quite similar among the samples.

The flavor attributes scores were quite different referred to bitter and sweet components. In this kind of food, sugar is one of the most important components, playing a key role in the structural and textural characteristics in dough arrangement and subsequent cooking, giving a distinctive shape and texture to the final product [49]. After the enrichment, the biscuits took on a more bitter taste, which means that the presence of these ingredients might have influenced the taste of the final products due to the presence of bitter-tasting compounds typically found in the fibrous component of the lemon peel. 

Finally, regarding the aromatic attributes, the enriched samples clearly differed from the control one, especially about the perception of the lemon scent. This suggests that the aromatic components in the fresh lemon peel and LP_E_ were preserved in the cooking process, enhancing the aroma in the final product. In addition, panelists did not detect the ethanol aroma in any biscuits.

## 4. Conclusions

The chemical composition of LP_E_ showed that it was a suitable source of polyphenols, and the obtained results remarked the positive effects of using lemon peel and its extract in biscuit-making.

Enriched biscuits showed higher polyphenol content and antioxidant activity than the control one, with suitable acceptability evaluated by the consumers through the preference test.

It may be concluded that lemon peel and LP_E_ could be incorporated in the formulation of biscuits, increasing their bioactive compounds content without affecting their overall appearance, resulting in acceptable products and with improved functional and nutraceutical properties. In this view, innovative uses of food wastes, rich in phenolics, can lead to an increase in the production of food with suitable nutritional characteristics and to satisfy customers’ expectations of healthy, functional foods.

## Figures and Tables

**Figure 1 foods-10-02460-f001:**
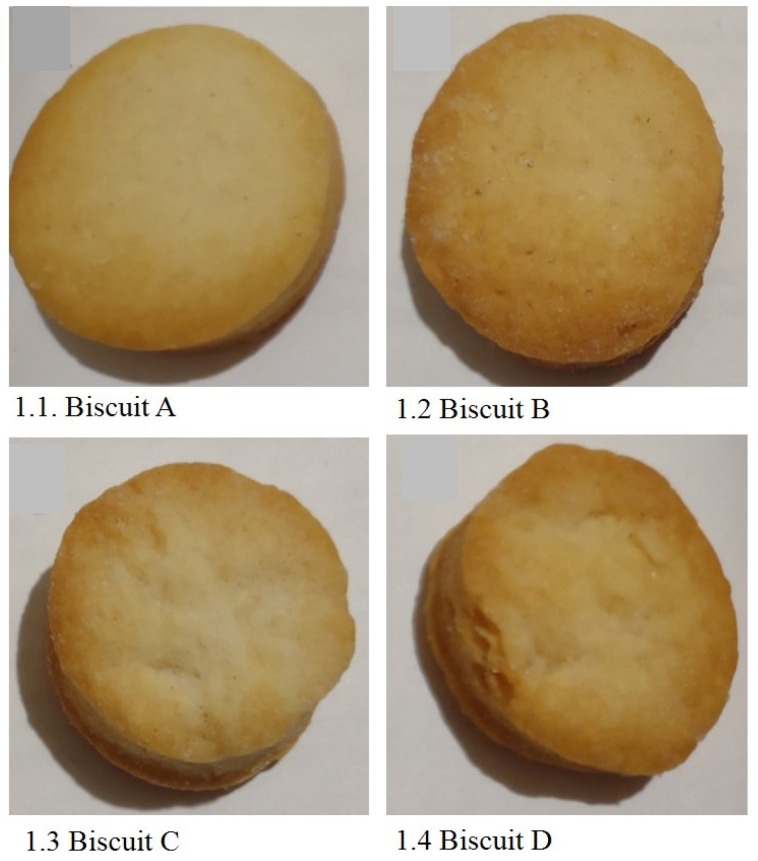
Image of the final product.

**Figure 2 foods-10-02460-f002:**
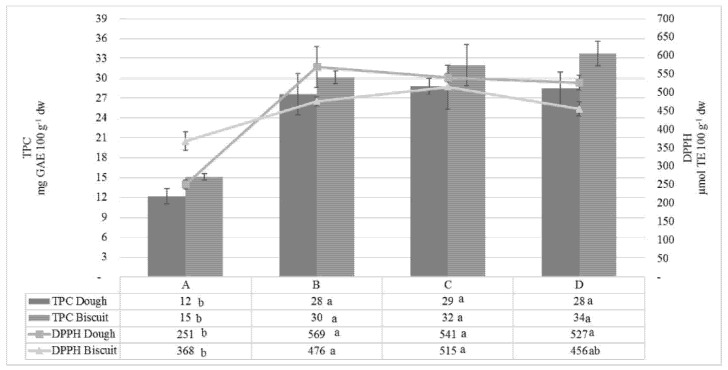
TPC values and antioxidant activity of dough and biscuit samples. Means within a row with different letters are significantly different by Tukey’s post-hoc test.

**Figure 3 foods-10-02460-f003:**
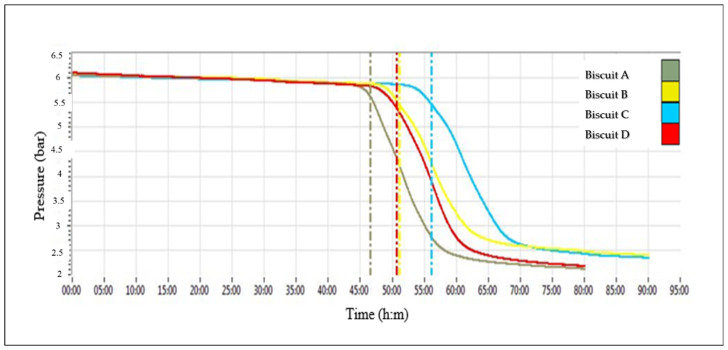
Oxidation curves and IP (induction period) values of biscuit samples.

**Figure 4 foods-10-02460-f004:**
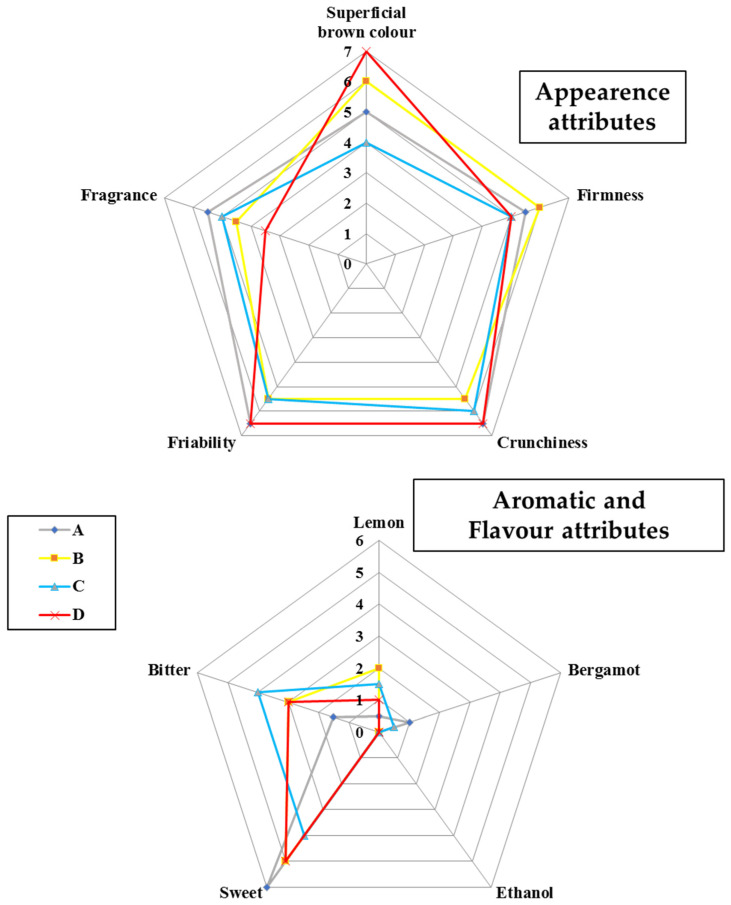
Median values of biscuits sensory evaluation.

**Table 1 foods-10-02460-t001:** Samples denomination.

A	Control			
B	Dough with fresh lemon peel			
C	Dough with fresh lemon peel and LP_E_			
D	Dough with LP_E_			
Ingredients	A	B	C	D
Wheat flour (g)	400	390	390	400
Corn seed oil (mL)	80	80	80	80
White sugar (g)	120	120	120	120
Skimmed milk (mL)	120	120	70	70
Baking powder (g)	8	8	8	8
Fresh lemon peel (g)	0	10	10	0
Lemon pomace extract (mL)	0	0	50	50

LP_E_: lemon peel extract.

**Table 2 foods-10-02460-t002:** Physicochemical characterization of LP_E_.

pH.	3.93 ± 0.01
Color:	L*: 42.33 ± 0.14
	a*: 1.22 ± 0.04
	b*: 7.12 ± 0.09
	C*: 7.22 ± 0.08
	H: 80.30 ± 0.38
TPC (mg GAE g^−1^ d.w.)	0.59 ± 0.01
TF (mg CE g^−1^ d.w.)	0.16 ± 0.00
DPPH (µmol TE g^−1^ d.w.)	1.86 ± 0.28
ABTS (µmol TE g^−1^ d.w.)	36.29 ± 4.62
Eriocitrin (mg 100 g^−1^ d.w.)	190.15 ± 0.04
Hesperidin (mg 100 g^−1^ d.w.)	231.60 ± 0.13

GAE: Gallic acid equivalent; TPC: Total phenolic Compouns; LP_E_ (lemon peel extract); TF: Total Flavonoids; DPPH and ABTS: Total antioxidant activity assays.

**Table 3 foods-10-02460-t003:** a_w_, moisture, and pH values of dough and biscuit samples.

DOUGH	a_w_	Moisture (%)	pH	BISCUIT	a_w_	Moisture (%)	pH
A	0.86 ± 0.00	19.28 ± 0.63 ^a^	7.13 ± 0.01 ^a^	A	0.28 ± 0.01 ^a^	4.69 ± 0.18	6.87 ± 0.03 ^a^
B	0.89 ± 0.00	18.70 ± 1.88 ^a^	7.15 ± 0.02 ^a^	B	0.28 ± 0.00 ^a^	4.76 ± 0.05	6.88 ± 0.00 ^a^
C	0.89 ± 0.00	18.25 ± 0.28 ^a^	6.91 ± 0.11 ^b^	C	0.28 ± 0.00 ^a^	5.16 ± 0.52	6.79 ± 0.02 ^b^
D	0.89 ± 0.00	14.14 ± 1.34 ^b^	6.87 ± 0.02 ^b^	D	0.23 ± 0.01 ^b^	4.60 ± 0.30	6.72 ± 0.03 ^c^
Sign	ns	**	**		**	ns	**

The data are presented as means ± SD (n = 3). Means within a column with different letters are significantly different by Tukey’s post-hoc test. Abbreviation: ns, not significant. ** Significance at *p* < 0.01.

**Table 4 foods-10-02460-t004:** Color parameters of dough and biscuit samples.

	Samples	A	B	C	D	Sign
Dough	L*	72.35 ± 2.08	72.09 ± 2.32	73.69 ± 2.39	70.98 ± 2.27	ns
	a*	2.50 ± 0.11 ^ab^	2.35 ± 0.17 ^bc^	2.24 ± 0.16 ^c^	2.57 ± 0.21 ^a^	**
	b*	18.90 ± 0.64 ^b^	19.74 ± 0.93 ^ab^	20.43 ± 0.92 ^a^	18.97 ± 0.38 ^b^	**
	C*	19.06 ± 0.64 ^b^	19.88 ± 0.94 ^ab^	20.55 ± 0.92 ^a^	19.15 ± 0.37 ^b^	**
	H	82.52 ± 0.36 ^b^	83.25 ± 0.46 ^a^	83.79 ± 0.44 ^a^	82.33 ± 0.65 ^b^	**
Biscuit	L*	65.39 ± 7.88	66.94 ± 5.91	68.99 ± 6.00	69.73 ± 3.32	ns
	a*	8.78 ± 3.71	8.96 ± 2.69	8.04 ± 3.31	8.15 ± 1.78	ns
	b*	27.56 ± 1.88	28.58 ± 1.41	27.28 ± 1.92	28.66 ± 1.24	ns
	C*	29.13 ± 2.03	30.07 ± 1.22	28.58 ± 2.45	29.83 ± 1.57	ns
	H	72.59 ± 0.13	72.63 ± 0.09	73.97 ± 0.10	74.27 ± 0.05	ns

The data are presented as means ± SD (n = 10). Means within a row with different letters are significantly different by Tukey’s post-hoc test. Abbreviation: ns, not significant. ** Significance at *p* < 0.01.

**Table 5 foods-10-02460-t005:** Total bacterial count (TBC) and mold (M) and yeast (Y) growth values for doughs.

DOUGH	TBC	M and Y
	(Log _10_ UFC g^−1^)	(Log _10_ UFC g^−1^)
A	2.22 ± 0.01 ^a^	1.76 ± 0.12
B	2.13 ± 0.12 ^a^	1.62 ± 0.03
C	2.10 ± 0.03 ^a^	1.45 ± 0.21
D	1.95 ± 0.02 ^b^	1.53 ± 0.24
Sign	**	ns

The data are presented as means ± SD (n = 3). Means within a column with different letters are significantly different by Tukey’s post-hoc test. Abbreviation: ns, not significant. ** Significance at *p* < 0.01.

**Table 6 foods-10-02460-t006:** Phenolic characterization of dough and biscuit samples. Data are expressed as mg 100 g^−1^ of product.

	Dough				Biscuit			
	B	C	D	Sign	B	C	D	Sign
Eriocitrin	3.54 ± 0.07 ^a^	3.40 ± 0.13 ^a^	1.62 ± 0.1 ^b^	**	nd	2.75 ± 0.16 ^a^	1.32 ± 0.20 ^b^	**
Hesperidin	2.32 ± 0.25 ^b^	2.67 ± 0.06 ^a^	2.41 ± 0.03 ^ab^	*	nd	2.84 ± 0.30 ^a^	2.22 ± 0.08 ^b^	**

The data are presented as means ± SD (n = 3). Means within a row with different letters are significantly different by Tukey’s post-hoc test. Abbreviation: ns, not significant; nd, not detected. ** Significance at *p* < 0.01; * significance at *p* < 0.05.

## Data Availability

Not applicable.

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
