# Peer review of "Functionalized Biscuits with Bioactive Ingredients Obtained by Citrus Lemon Pomace"

_foods, 2021, doi:10.3390/foods10102460_

Round 1
Reviewer 1 Report
All the changes have been addressed
Author Response
We are grateful to the Reviewer for your positive reply to our paper.
Reviewer 2 Report
The manuscript reports on work incorporating lemon pomace extracts into biscuits to enhance antioxidant activity and potentially shelf life of the products. The general study design and methodologies are appropriate, results appear to be sound. I would recommend for consideration with minor revision. See below.
Line 82-83 - 50% EtOH extract was used to prepare biscuits - is the EtOH food grade? The wording for human consumption should be revised, as the liquid extract was not directly consumed as the ingredients mixture will go further processing and dried into biscuits.
Line 156 - what is the concentration of chloridric acid (suggest to use hydrochloric acid)?
Line 200-202 - The DPPH antioxidant result should be expressed as µmol TE, not µM TE. Same issue with ABTS.
Author Response
The article has been carefully reviewed and the changes requested by the reviewer have been accepted.
1) Line 82-83- The extraction solvent was food-grade ethanol. As the reviewer advised the sentence (human consumption) was deleted and modified in the text.
2) Line 156- for the analysis concentrated hydrochloric acid (37%) was used. The sentence was modified in the text.
Line 200-202- the unit of measure has been changed
This manuscript is a resubmission of an earlier submission. The following is a list of the peer review reports and author responses from that submission.
Round 1
Reviewer 1 Report
The manuscript contains useful information on the use of lemon pomace extract to elaborate biscuits to obtain healthier bakery products. The pertinent analyses have been conducted and the results are interesting. However, there are some suggestions and changes that should be made. They are pointed out below.
Introduction:
- Line 32: “processed”. Info about the type of processing should be included: juices, jams??
- Line 54. realizing? Better preparing.
Material and methods:
- Line 63 “freezer-dried”. Give equipment and conditions, please.
- Line 66. please give type of baking powder
- Line 75. “the extraction solvent”. Is this extraction solvent ethanol: water 1:1? Is this solvent suitable for human consumption? Please, give some references of using ethanol extracts for food enrichment.
- Line 81-82. “lemon peels”, which lemon peels? Fresh lemon peels ( nothing to do with lemon pomace)? Please, explain in the manuscript
- Line 83-84. “electric mixer”. Device and conditions should be included.
- Lines 85-87. Do you sheet the dough? How do you assure the same thickness for all the samples?
- Line 89. “subsequent,”?? Perhaps the comma is not needed here
- Section 2.4.1. C* and h* would be better parameters to explain colour changes.
Results and Discussion:
- Line 235: “Water content or water activity (aw)”??
- Line 236. Which thermal treatment? Baking in the oven? If so, please delete (biscuits are always thermal treated....)
- Line 237. free water or water activity (aw)? Please, keep consistency in the terms used.
- Line 241-242. Please, explain the results using the correct statistic terms. Here: "Dough D showed the lowest value (p<0.05) of moisture ....."
- Table caption. "row" or "column". Check all the table captions, please.
- Line 259. “especially for b* that increased in B and C,”. Please, calculate C* and h*. this parameters will explain better than b* the changes to "yellow or red colour".
- TPC results. The authors should consider the loss of water during baking, as they are giving the results of TPC in wet basis. So, I am not sure if it makes sense to compare doughs with biscuits.
- Figure 1 caption. Try to keep consistency. You should add: "Means within a row ( or column) with different letters are significantly different by Tukey's post hoc test"
- Line 355. "discussed below"? Better "observed in Figure 2"
- Figure 2 should be redone, using different colours or different types of line to distinguish among samples. and please, correct the typo "Bisciut A"
- Line 369. “useful compounds"?? What do the authors mean here?
- Line 384. “"superficial brown colour" instead of " superficial color"
- Line 387. Please, check this typo throughout the manuscript.
- Line 388. “crispiest" or "crunchiest". Crispy and crunchy are not synonyms when speaking in sensory terms.
- Lines 400-401. Please delete this sentence, as a liking test has not been performed.
- Moreover, "taste" should be removed from the graphs or better explained. Which taste were the panelists asked for? A specific taste attribute should have been scored and "taste" is a general one that cannot be scored (all the foods have taste...)
Conclusions:
Line 419-421. “Please, remove this sentence. It is not
Reviewer 2 Report
Introduction: authors do not give the chemical composition of citrus lemon pomace (quantitative and qualitative composition in phenolic compounds, citric acid, sugars, water, ascorbic acid, etc.).
Line 30. I would write “most studies are focused”.
Line 46. I would write “good/interesting nutritional qualities”.
Materials and methods. Authors do not specify if citrus lemon pomace was organic. If not, they do not talk about the residual pesticides on these kind of wastes that are harmful for human consumption. Residual pesticides in agricultural wastes pass to food, in this case, to biscuits.
Line 78. Erase “(LPE)”
Paragraph 2.3. I think that authors should have add all ingredients at the same proportion and use water as the “replacing” ingredient to compensate the addition of lemon pomace extract (containing water). In this case, the total water amount in the formulation would have been the same in ever sample. More importantly, the same amount of molecules possessing functional properties in the base ingredients (flour, oil, sugar, milk, baking powder), would have been the same. Authors decreased skimmed milk to compensate the addition of water in the lemon pomace extract in samples C and D. However, skimmed milk contain molecules that influence the overall antioxidant and organoleptic quality of dough and biscuits and this is not mentioned, nor discussed, anywhere in the article. For example, milk’s proteins possess emulsifying/gelling properties that can influence the final texture of dough and biscuits, milk’s peptides can present antioxidant activity, lactose’s and protein’s milk participate to Maillard Reaction, and lactose can also caramelize in the surface of biscuits during cooking. So direct comparison of samples A and B with C and D cannot be made, as in samples C and D, functional molecules of milk are not present at the same concentration.
No chemical characterization of lemon peel (contents in vitamin C, reducing sugars, phenolic compounds, etc.) is presented. For the lemon pomace extract neither dry matter nor vitamin C contents are reported in Table 2. It is not clear if results in table 2 are in dry or wet basis, thus it is difficult to calculate for example the theoretical total eriocitrin or hesperidin added in formulations C and D, as in table 1 the lemon pomace extract is in mL. Authors did not compare the theoretical total phenol, eriocitrin and hesperidin contents added to samples versus the quantified ones. This information could help to see if the phenol compounds extraction method form dough and biscuit was efficient, and also to prove if an additive/antagonist/synergist effect was provoked when adding lemon peel and/or lemon pomace extract.
I would have added an analysis of vitamin C for all samples. No discussion about the influence of vitamin C neither on the antioxidant activity nor the organoleptic (texture, the known effect of ascorbic acid on gluten) is presented on the article.
Line 104. Where were put the 15 mL of LPE for the color analysis? In a Petri plate with a black bottom?
Total flavonoid and phenolic compounds results are expressed in mg/g. They did not specify if is on dry or wet basis. In 2.4.3. they take 5 g of ground dough or biscuit. How did they grind samples? Did they freeze/freeze-dried dough?
Line 132-134. I would write “mg of gallic acid equivalent”, “mg GAE g-1 d.b. or w.b.?”)
Lines 147-148. Authors talk about mg of catechin equivalents (d.b. or w.b.?).
No description of the method to identify/quantify eriocitrin and hesperidin is present.
Line 237. Aw is different of DM (dry matter).
How do you explain no significant differences in L a* b* results, when adding more skimmed milk (containing proteins+lactose) on samples A-B versus C-D?
Which was the influence of pH on the microorganism growth?
I think that TBC, M and Y should have been determined also on lemon peel and lemon pomace extract, as they modified the amount of skimmed milk on samples C and D. The lower bacterial count in sample D could be related to the lower skimmed milk content?
The kind of bacteria, moulds and yeast brought by milk and by on lemon peel and lemon pomace extract can be different. That is why is important to add the same quantity of skimmed milk to each formulation. Milk has also a buffer action and lemon peel and lemon pomace extract contain citric acid; all this influence microorganisms’ development, which was not discussed.
Lines310-313. Explanation about no significant differences is not clear. In B, C, and D samples the TPC content is significantly higher in biscuit than in dough than in the control (A), and not only D as written on line 312. Increases go from 9 to 19 %. But according to results in the table in figure 1, no significant differences were found. If I well understood, the Anova test was made to compare results in arrows, ex. TPC dough results for A, B, C, and D. Different signs should have been used to compare TPC dough versus TPC biscuit and DPPH dough versus DPPH biscuit for each sample.
Figure 1. ** is not defined. TPC and DPPH results are expressed as dry basis?
How do you explain that the TPC, DPPH, and eriocitrin/hesperidin contents are not always significantly higher in sample C? In theory sample C contains more natural antioxidant (vitamin C + phenolic compounds) than the other samples.
Lines 343-346; References are not given about the reported synergistic antioxidant effect exerted by the added lemon peel and LPE. The synergistic mechanism is not explained, the effect of vitamin C is not mentionned, the synergistic effect of vitamin C-phenolic compounds neither.
Line 361-362. According to table, the content in dough on eriocitrin on sample C is not significantly superior to B, and on hesperidin is not significantly superior to D. the sentence “C dough and biscuit…. Enriched samples” is not exactly.
Paragraph 3.6. As in lines 361-362, discussion of results is not clear, comparisons are not well made in comparison with the Anova analysis. C>B in hesperidine dough. C>D in eriocitrin and hesperidin biscuit.
Why eriocitrin and hesperidin total amounts are not significantly superior on sample C in dough and biscuit? If lemon peel and LPE contain these compounds? Which is the content in these compounds in lemon peel?
An important result was not well discussed: why no eriocitrin nor hesperidin were detected on biscuit B. A problem with the extraction method? The thermal effect was supposed to be the same for all samples, unless they did not respect the same temperature and time in the oven. So lines 373-375 are not a valuable explanation.
Paragraph 3.7.
Line 385. Sample C contains less skimmed milk, thus less proteins and lactose, thus less Maillard reaction and caramelization.
Line 386. Saccharose can also caramelize when its melting point is reached, not only reducing sugars. Saccharose was added to the formulation and it is not a reducing sugar. Reducing sugars as lactose in milk participates to Maillard reaction. That is why sample A had more superficial brown colour. pH also influences on Maillard reaction and caramelization derived products. You should discuss this part also.
Some photos of the final biscuits to compare the color would have been appreciated.
Line 387. “was quite similar… except for sample D”. How do you explain this?
In this part there is no discussion about the effect of vitamin C, phenolic compounds, milk proteins… on texture from the mechanistic/chemical point of view. It is well known that ascorbic acid has an effect on gluten, milk proteins have emulsifying properties that influence the texture of the final product, milk constituents influences also the flavor, color and aroma.
Conclusion. Lines 419-421. This part should be placed and more developed on discussion. No comments about the possible presence and harmfulness on lemon peel and LPE of pesticides and on the importance of using wastes derived from organic agriculture, appear neither in discussion nor in conclusion.